# Spent Coffee Grounds Alter Bacterial Communities in Latxa Dairy Ewes

**DOI:** 10.3390/microorganisms8121961

**Published:** 2020-12-10

**Authors:** Idoia Goiri, Xabier Díaz de Otálora, Roberto Ruiz, Jagoba Rey, Raquel Atxaerandio, Jose Luis Lavín, David San Martin, Mikel Orive, Bruno Iñarra, Jaime Zufia, Jabi Urkiza, Aser García-Rodríguez

**Affiliations:** 1Department of Animal Production, NEIKER-Basque Institute for Agricultural Research and Development, Basque Research and Technology Alliance (BRTA), Campus Agroalimentario de Arkaute s/n, 01192 Arkaute, Spain; xabierdiazdeotalora@gmail.com (X.D.d.O.); rruiz@neiker.eus (R.R.); jrey@neiker.eus (J.R.); ratxaerandio@neiker.eus (R.A.); jllavin@neiker.eus (J.L.L.); aserg@neiker.eus (A.G.-R.); 2AZTI, Food Research, Basque Research and Technology Alliance (BRTA), Parque Tecnológico de Bizkaia, Astondo Bidea, Edificio 609, 48160 Derio-Bizkaia, Spain; dsanmartin@azti.es (D.S.M.); morive@azti.es (M.O.); binarra@azti.es (B.I.); jzufia@azti.es (J.Z.); 3Cooperativa Agraria MIBA, Polígono Industrial Galartza, 48277 Etxebarria, Spain; jabi@miba.coop

**Keywords:** circular economy, coffee by-products, ruminal microbes

## Abstract

Antimicrobial and antioxidant properties of spent coffee grounds (SCG) make them a potential ingredient in a diet for ruminants. This study investigated the effects of SCG on rumen microbiota. For 51 days, 36 dairy ewes were assigned to the experimental treatments (0, 30, 50, and 100 g SCG/kg). Ruminal samples were collected on day 50. DNA was extracted and subjected to paired-end Illumina sequencing of the V3-V4 hypervariable region of the 16S rRNA genes. Bioinformatic analyses were performed using QIIME (v.1.9.0). SCG increased dose-dependently bacterial diversity and altered bacterial structure. Further, 60, 78, and 449 operational taxonomic unit (OUT) were different between control and 30, 50 and 100 g/kg SCG groups, respectively. Higher differences were observed between the control and 100 g/kg SCG group, where OTU of the genera *Treponema*, *CF231*, *Butyrivibrio*, *BF331*, *Anaeroplasma*, *Blautia*, *Fibrobacter*, and *Clostridium* were enriched with SCG. Correlations between volatile fatty acids (VFA) and bacterial taxa were sparser in the SCG groups and had little overlap. Certain bacterial taxa presented different signs of the correlation with VFA in SCG and control groups, but *Butyrivibrio* and *Blautia* consistently correlated with branched-chain VFA in all groups. SCG induced shifts in the ruminal bacterial community and altered the correlation networks among bacterial taxa and ruminal VFA.

## 1. Introduction

Coffee is one of the most valuable primary products in world trade due to the high consumption of coffee beverages [1]. According to the International Coffee Organization statistics, world coffee production in 2019 was 10.13 million tons, of which EU countries consume about 3.3 million tons. Coffee consumption leads to amounts of organic waste; the primary by-product of coffee production is spent coffee grounds (SCG).

SCG contain a wide range of components formed through the Maillard reactions during the roast, such as melanoidins [2] and polyphenols [3], which are supposed to have antimicrobial properties. For instance, polyphenols like those present in SCG are known to interact with rumen microbiota, affecting rumen fermentation and even lipid metabolism by depressing the biohydrogenation of unsaturated fatty acids (UFA) [4]. As a consequence, these by-products have been used as functional ingredients in dairy sheep formulations [5]. According to these authors, the inclusion of SCG up to 100 g/kg in the concentrate modified the fermentation pattern towards increased branched-chain volatile fatty acids (BCVFA), acetic and butyric acid proportions in the rumen resulting in improvements of milk yields and quality. Isoacids are essential nutrients for certain rumen microorganisms, and have been reported to enhance the growth of fiber-digesting microorganisms in the rumen. Thus, it could be hypothesized that the observed greater proportion of isoacids in the rumen in ewes fed SCG could lead to a change in rumen bacterial communities which could alter the rumen fermentation pattern and explain the observed shift towards greater acetic and butyric acid proportions [6]. To the best of the authors knowledge, the effects of SCG on rumen microbial populations have not been reported in the literature. Therefore, the current study was performed as an accompaniment to [5], based on the hypothesis that the observed shift towards a greater acetic and butyric acid content is modulated by alterations in the ruminal bacterial community.

To test the above hypothesis, this study was performed to investigate the effects of SCG at four doses with practical and productive relevance to the composition and structure of rumen microbiota and their associations with rumen fermentation parameters.

## 2. Materials and Methods

All experimental procedures were performed in accordance with the European Union Directive (2010/63/EU) and Spanish Royal Decree (RD 53/2013) for the protection of animals used for experimental and other scientific purposes, and approved by the ethics committee (NEIKER-OEBA-2018-004, 15 January 2018).

### 2.1. Spent Coffee Grounds Collection and Drying

This study was focused on the SCG produced by hotels, restaurants and coffee shops located in the north of Spain (Basque Country and Navarre) and south of France (Aquitaine). About 0.5 tons of SCG were stored and collected during a week. This amount of SCG was transported to a drying plant for its processing as an ingredient for dairy ewe feed. SCG were stabilized and dried using the flash dryer technology (RINA-JET 1008, Riera Nadeu S.A., Barcelona, Spain) as described in [7].

### 2.2. Experimental Design, Animals and Feeding

The design of the experiment and animal assignment to treatments was previously described in [5]. Briefly, 36 lactating Latxa ewes were used in a randomized block design and were equally distributed into four experimental groups that either received no SCG (0; *n* = 9), 30 g/kg dry matter (DM) SCG/d (30; *n* = 9), 50 g/kg DM SCG/d (50; *n* = 9), and 100 g/kg DM SCG/d (100; *n* = 9). Ewes were blocked according to initial milk production and days in milk. All sheep with the same dietary treatment were accommodated in the same pen. Based on a requirement for 2000 g milk/d all sheep were fed a pelleted concentrate (950 g DM/d) and ad libitum fescue hay (Festuca pratensis) for 51 days (Hassoun and Bocquier, 2007). The concentrate was supplied in individual feeders in the milking parlour during the morning (7:30) and afternoon (18:00) milkings. Fescue hay (without chopping) was group fed in a feed bunk (0.5 m/ewe). The quantity of fescue hay offered was based on morning bunk readings, and the amount of feed offered was adjusted daily to allow 10% refusals. Feed samples (hay and concentrate) were collected for each treatment for chemical analyses. Details of the experimental diets are presented in (Appendix A)

### 2.3. Samplings and Measurements

#### 2.3.1. Sampling of Ruminal Contents

On the last day of the trial, after the previous afternoon milking ewes were fasted. Ruminal samples were collected from each dairy ewe after the morning milking using an esophageal tube (0.9 cm in diameter and 150 cm in length) connected to a mechanical pumping unit (Vacuubrand ME 2SI, Wertheim, Germany). Ruminal samples were filtered through four layers of sterile cheese cloths, and then about 10 mL of each ruminal extraction were placed into a container and were immediately frozen and stored at −20 ± 5 °C until further analysis of volatile fatty acids (VFA). Samples of ruminal contents were also collected and immediately frozen and conserved at −20 ± 5 °C for DNA extraction for bacterial community studies.

#### 2.3.2. DNA Extraction and Illumina Library Generation

The DNA extraction of rumen liquid samples was performed using the commercial Power Soil DNA Isolation kit (Mo Bio Laboratories Inc., Carlsbad, CA, USA) following the manufacturer’s instructions. The integrity of the DNA was checked on 0.8% (wt/vol) agarose gels and the yield and purity of extracted DNA were determined using a NanoDrop spectrophotometer (NanoDrop^®^ND-1000, Thermo Fisher Scientific, Waltham, MA, USA) checking the 260/280 nm ratio. The purity of DNA was considered acceptable with ratios between 1.8–2.0. The extracted DNA was subjected to paired-end Illumina sequencing of the V3-V4 hypervariable region of the 16S rRNA genes [8]. Libraries were generated using Illumina’s Nextera kit. The 300bp paired-end sequencing reactions were performed on a MiSeq platform (Illumina, San Diego, CA, USA). Read’s quality control was performed via FASTQC software, then forward and reverse reads were merged using FLASH-1.2.11 [9] with 10 bp minimum overlap and allowing “outie” orientation. Operational taxonomical unit (OTU) detection, taxonomy determination, phylogeny inference and OTU tables generation were done by the open-source software package QIIME (v.1.9.0): Quantitative Insights into Microbial Ecology software package [10]. The sequences were clustered as OTUs of 97% similarity using UCLUST [11]. The OTUs were checked for chimeras using the RDP gold data-base and assigned taxonomy using the Greengenes database [12]. The alpha and beta diversity metrics were also calculated by the QIIME pipeline.

The sequence data has been deposited in the European Nucleotide Archive database under the accession number PRJEB41182.

### 2.4. Chemical Analyses

Roughage and concentrate were dried in a forced-air oven. All samples including SCG were ground to pass a 1-mm screen and analysed following official methods [13]. DM content (method 934.01) and ash (method 942.05) were determined. Nitrogen content (method 990.03) was determined using the macro-Kjeldahl procedure on a Kjeltec Auto 1030 (Foss, Hillerød, Denmark). Neutral detergent fibre (NDF) was determined with use of an alpha amylase, but without sodium sulphite, and was expressed free of ash [14]. Acid detergent fibre (ADF) was determined and expressed exclusive of residual ash [15]. Fat content was determined (method 920.39) without hydrolysis by the automated soxhlet method (Selecta S.A., Barcelona, Spain) using hexane for 6 h as solvent. Starch content was measured by polarimetry [16]. Concentrates were analysed for caffeine [17] total phenolic compounds [18] and melanoidins [19].

The analysis of VFA (acetic, propionic, butyric, isobutyric, valeric and isovaleric) of rumen samples was performed by gas chromatography as described in [5]. Quantification expressed in mmol/L was done using an external calibration curve based on standards. Data were expressed in mmol/mol.

### 2.5. Calculations and Statistical Analysis

Branched-chain volatile fatty acid values were calculated as isovaleric plus isobutyric acids.

For the statistical analysis, each dairy ewe (*n* = 36) was considered as the experimental unit.

Relative abundances of bacterial taxa at phylum, family and genus level were analyzed using the GLM procedure (SAS, 2017), according to the following model:Y_jk = μ + T_j + B_k + ε_jk
where Y is the dependent variable, μ is the mean values for each treatment, T is the fixed effect of the concentrate used, B is the fixed effect of the block, and ε are the residuals.

Due to the unequal doses, spaced coefficients for orthogonal polynomials were calculated using the ORPOL function in PROC IML in SAS [20], to determine linear and quadratic trends.

Residuals were checked for normality using either Shapiro-Wilk or Kolmogorov-Smirnov test. When residuals did not follow a normal distribution, data were transformed (ln, square-root and reciprocal transformation) to follow a normal distribution.

Statistically significant differences between experimental groups’ bacterial community composition, were tested by analysis of dissimilarity (ADONIS) with 999 permutations. The significant fold changes of the OTU’s were tested by DESeq2 [21] and filtered by false discovery rate (FDR) value.

To investigate the correlations between the ruminal VFA and bacterial taxa, a regularized canonical correlation analysis (rCCA) was performed using the package mixOmics (v6.6.2) [22] in R (v3.5.1) [23]. To perform the rCCA analysis, the correlation values between the relative abundances of bacterial taxa (at genus level) and each ruminal VFA proportions were computed to calculate a similarity matrix. A clustered image map was inferred using a similarity matrix obtained from the rCCA. The relevant components were obtained setting a threshold to R = 0.45.

## 3. Results

### 3.1. Ruminal Bacterial Community

After a preliminary analysis of sequenced data one sample of the control group was identified as a technical outlier and discarded. Figure 1 represents the bacterial community composition at the family level in the ruminal samples of sheep when fed the different dietary treatments. The most abundant phyla were Bacteroidetes (68.8%) and Firmicutes (24.8%), followed by Spirochaetes (2.5%), Proteobacteria (1.0%), Verrucomicrobia (0.8%) and Lentisphaerae (0.7%). Within Bacteroidetes, the dominant families in order of importance were Prevotellaceae (predominant genus *Prevotella*: 52.8%), undefined families within the order of the Bacteroidales (10.8%) and (Paraprevotellaceae) (3.0%). The predominant families of Firmicutes were Lachnospiraceae (9.5%), Ruminococcaceae (8.3%) and undefined families within order of the Clostridiales (5.7%), whereas Spirochaetes consisted mainly of family Spirochaetaceae (predominant genus *Treponema*: 2.4%). The predominant families of Proteobacteria were Desulfovibrionaceae (0.3%) and Xanthomonadaceae (0.2%), whereas the predominant family for Verrucomicrobia and Lentisphaerae were RFP12 (0.7%) and Victivallaceae (0.7%), respectively.

Supplementation of concentrate with spent coffee grounds affected in a different manner richness and diversity indexes (Table 1). The number of observed OTU (*p* > 0.10) and Chao 1 richness estimate (*p* = 0.069) were not affected by SCG, but phylogenetic diversity (*p* = 0.007) and Shannon index (*p* = 0.001) increased linearly with SCG dose in the concentrate.

Moreover, there were community structure shifts caused by the inclusion of SCG in the concentrate of dairy sheep. The beta diversity analysis showed a clustering of the treatments. Control and concentrate with 100 g/kg of SCG clustered separately in the PCoA plot, while treatment with 30 and 50 g/kg of SCG appeared between both of them (Figure 2). The statistical test performed with ADONIS revealed differences (*p* = 0.001) in bacterial community structure between experimental concentrates.

Thirteen different phyla were identified in all the samples (Appendix A). Among them, Cyanobacteria (linearly decreased; *p* < 0.001), Firmicutes (linearly increased; *p* = 0.038), Lentisphaerae (linearly decreased; *p* = 0.003), Proteobacteria (linearly decreased; *p* = 0.011), Spirochaetes (tended to increase linearly; *p* = 0.07), Tenericutes (linearly increased; *p* = 0.023), and Verrucomicrobia (linearly decreased; *p* < 0.001) were influenced by the SCG treatments. One hundred and twenty-nine different families were found from at least one sample, and 21 families including 2 unclassified families (UF) had a relative abundance of >0.5% in at least one of the samples (Appendix A). Among all these major families, UF within the order Bacteroidales (*p* = 0.009), BS11 (*p* < 0.001), Bacteroidaceae (*p* = 0.016), UF within the order Clostridiales (*p* = 0.001), Ruminococcaceae (*p* = 0.001), Anaeroplasmataceae (*p* = 0.02), and Spirochaetaceae (*p* = 0.06) increased or tended to increase linearly and Prevotellaceae (*p* = 0.004), Victivallaceae (*p* = 0.004), and RFP12 (*p* < 0.001) linearly decreased with increased doses of SCG. In contrast, RF16 (*p* = 0.03), Veillonellaceae (*p* = 0.05), Mogibacteriaceae (*p* = 0.01), Xanthomonadaceae (*p* = 0.01) were quadratically affected.

The present study identified 242 genera in at least one sample. Twenty-seven genera had relative abundances >0.5% (Appendix A) Among them, undefined genera (UG) within the order Bacteroidales (*p* = 0.006), UG within the order Clostridiales (*p* = 0.001), UG within family Ruminococcaceae (*p* = 0.002) UG within family BS11(*p* < 0.001), *BF311* (*p* = 0.02), *Treponema* (*p* = 0.062) linearly increased or tended to increase and *Prevotella* (*p* = 0.004), *Anaerovibrio* (*p* = 0.004), *Selenomonas* (*p* < 0.001), UG within family Victivallaceae (*p* = 0.004), UG within family RFP12 (*p* < 0.001) linearly decreased with increased doses of SCG. In contrast, UG within family RF16 (*p* = 0.03), *CF231* (*p* = 0.04) were quadratically affected.

Indeed, 7034 operational taxonomic unit (OTU) were shared among all the treatments, but 268, 247, 279, and 344 OTU were unique to the control, 30 g/kg SCG, 50 g/kg SCG, and 100 g/kg SCG, respectively (Figure 3).

DESeq2 analysis showed that there were 60 OTU and 78 OTU significantly different in abundance between control and 30 g/kg and 50 g/kg SCG groups, respectively, while there were 449 OTU significantly different in abundance between control and 100 g/kg SCG. Spent coffee grounds groups shared 15 OTU significantly different with control, but 24, 25, 381 OTU significantly different with control were unique to 30 g/kg, 50 g/kg, and 100 g/kg SCG, respectively (Appendix A)

The OTU with significant differences between control and 30 g/kg SCG groups are shown in Appendix A. Some OTU of the genera *Prevotella*, *YRC22*, *Coprococcus*, *Desulfovibrio*, undefined genera (family Ruminococcaceae, (Paraprevotellaceae), BS11), and within order Bacteroidales were enriched in the samples of ruminal content of sheep fed a concentrate with 30 g/kg of SCG. On the other hand, in control samples, enrichment was observed in some OTU of the genera *Ochrobactrum*, *Prevotella*, *Sphingobacterium*, *Achromobacter*, *Dietzia*, *Rhodococcus*, *Aeromicrobium*, *Facklamia*, *Treponema*, *Renibacterium*, *Corynebacterium*, *Oscillospira*, *Ruminococcus*, and undefined genera (family Xanthomonadaceae, Alcaligenaceae, Acetobacteriaceae, and Lachnospiraceae)

The OTU with significant differences between control and 50 g/kg SCG groups are shown in Appendix A. Some OTU of the genera *Prevotella*, *YRC22*, *Coprococcus*, *Fibrobacter*, *Anaeroplasma*, undefined genera (family Ruminococcaceae, (Paraprevotellaceae)), and within order Clostridiales and Bacteroidales were enriched in the samples of ruminal content of sheep fed a concentrate with 50 g/kg of SCG. In control samples enrichment was observed in some OTU of the genera *Ochrobactrum*, *Prevotella*, *Sphingobacterium*, *Achromobacter*, *Dietzia*, *Rhodococcus*, *Pseudomonas*, *CF231*, *Stenotrophomonas*, *Aeromicrobium*, *Luteimonas*, *Renibacterium*, *Oscillospira*, *Selenomonas*, *Pseudobutiryvibrio*, *Succinivibrio*, and undefined genera (family Xanthomonadaceae, Alcaligenaceae, Acetobacteriaceae, Lachnospiraceae, Phyllobacteriaceae, Victivallaceae, Methylobacteriaceae, Caulobacteriaceae, (Mogibacteriaceae), Comamonadaceae).

The OTU with significant differences between control and 100 g/kg SCG groups are shown in Figure 4. Some OTU of the genera *Prevotella*, *YRC22*, *Ruminococcus*, *Coprococcus*, *Treponema*, *CF231*, *Butyrivibrio*, *BF331*, *Anaeroplasma*, *Blautia*, *Fibrobacter*, *Clostridium*, undefined genera (family BS11, Ruminococacceae, (Paraprevotellaceae), (Mogibacteriaceae), Lachmospiraceae, Christensenellaceae), and within order Bacteroidales and Clostridiales were enriched in the samples of ruminal content of sheep fed a concentrate with 100 g/kg of SCG. In control samples, enrichment was observed in some OTU of the genera *Ochrobactrum*, *Prevotella*, *Sphingobacterium*, *Dietzia*, *Aeromicrobium*, *Achromobacter*, *Luteimonas*, *Selenomonas*, *Rhodococcus*, *Corynebacterium*, *Delftia*, *YRC22*, *Stenotrophomonas*, *Oscillospira*, *Pseudomonas*, *Staphylococcus*, *Anaerovibrio*, *Succinivibrio*, *Ruminococcus*, *Coprococcus*, *Pseudobutiryvibrio*, *Devosia*, *Renibacterium*, *Treponema*, *CF231*, undefined genera (family Xanthomonadaceae, Alcaligenaceae, Phyllobacteriaceae, Acetobacteriaceae, RFP12, Victivallaceae, Lachnospiraceae, WCHB1-25, Coriobacteriaceae, Methylobacteriaceae, Comamonadaceae, Ruminococacceae, (Paraprevotellaceae), Caulobacteriaceae), and within order Rhizobiales, Bacteroidales, Clostridiales, and RF32.

### 3.2. Correlations between Bacterial Taxa and Ruminal Volatile Fatty Acids

The correlations between ruminal VFA proportions and bacterial taxa were represented by a clustered image map inferred from the rCCA analysis. Irrespective of the treatment (CTR or SCG at 30, 50 and 100 g/kg; Figure 5), 22 different OTU were associated with ruminal VFA taking into account the established cutoff of 0.45. Genera *Blautia*, *CF231*, (*Prevotella*), *Butyrivibrio*, *Mogibacterium*, *Moryella*, undefined genera within families Ruminococacceae, BS11, Christensenellaceae, (Mogibacteriaceae), Clostridiaceae, S24-7, Desulfovibrionaceae, Lachnospiraceae, and order Clostridiales were positively correlated with isovaleric, isobutyric and total BCVFA proportions, whereas genera *Coprococcus*, *Prevotella* and undefined genera within family RF16 were negatively correlated with them. Genera *Prevotella*, *Oscillospira*, *Selenomonas*, and undefined genera within family WCHB1-25 were positively correlated, whereas genera *BF311* and undefined genera within families Ruminococcaceae, BS11, Christensenellaceae, and order Clostridiales were negatively correlated with propionate proportion. Regarding acetate molar proportions, genera *BF311* and undefined genera within families BS11, Ruminococcaceae, and within order Clostridiales were positively correlated whereas genera *Prevotella*, *Oscilospira*, and undefined genera within family WCHB1-25 were negatively correlated.

In the control group (Figure 6A), a higher number of OTU (47) were associated with ruminal VFA proportions taking into account the same cutoff of 0.45. Genera *Blvii28*, *Prevotella*, *Mogibacterium*, *Ochrobactrum*, *Blautia*, *Pseudomonas*, *Succinivibrio*, *Butyrivibrio*, *Streptococcus*, *Anaerovibrio*, *Ruminobacter*, *Desulfovibrio*, *Ruminococcus*, *Pseudobutyrivibrio*, *CF231*, (*Prevotella*), *Succiniclasticum*, *Moryella*, undefined genera within families Christensenellaceae, Clostridiaceae, WCHB1-25, Desulfovibrionaceae, Succinivibrionaceae, Victivallaceae, Xanthomonadaceae, Veillonaceae, S24-7 and within order RF32 were positively correlated with ruminal proportions of isobutyric, isovaleric and total BCVFA, whereas genera *Coprococcus*, *Anaeroplasma*, *Clostridium*, *YRC22*, *Fibrobacter*, *RFN20*, *BF311*, *Shuttleworthia*, *Treponema*, and undefined genera within families RF16, (Paraprevotellaceae), BS11, Spirochaetaceae, and within orders Clostridiales, Bacteroidales, Rickettsiales and ML615J-28 were negatively correlated with them. Genera *Treponema*, *Blvii28*, *Prevotella*, *Mogibacterium*, and undefined genera within families Christensenellaceae, Clostridiaceae, WCHB1-25, Desulfovibrionaceae and Succinivibrionaceae were positively correlated, whereas genera *Ochrobactrum*, *Blautia*, *Pseudomonas*, *Succinivibrio*, *Ruminobacter*, *Selenomonas*, *Pseudobutiryvibrio*, and undefined genera within families Coriobacteriaceae, Victivallaceae, Xanthomonadaceae, Veillonellaceae, and within order RF32 were negatively correlated with propionate. Regarding acetate ruminal proportions, genera *Coprococcus*, *Anaeroplasma*, *Clostridium*, *YRC22*, *Fibrobacter*, *RFN20*, *Ochrobactrum*, *Blautia*, *Pseudomonas*, *Succinivibrio*, *Ruminobacter*, *Selenomonas*, *Pseudobutiryvibrio*, *CF231*, (*Prevotella*) and undefined genera within families RF16, (Paraprevotellaceae), BS11, Spirochaetaceae, Coriobacteriaceae, Victivallaceae, Xanthomonadaceae, Veillonellaceae and within orders of Clostridiales, Bacteroidales, Rickettsiales and RF32 were positively correlated, whereas genera *Treponema*, *Blvii28*, *Prevotella*, *Mogibacterium*, and undefined genera within families Christensenellaceae, Clostridiaceae, WCHB1-25, Desulfovibrionaceae, and Succinivibrionaceae were negatively correlated.

In the groups fed with SCG at different doses, 22 different OTU were associated with ruminal VFA proportions taking into account the established cutoff of 0.45 (Figure 6B). Genera CF231, *Pseudomonas*, (*Prevotella*), *Butyrivibrio*, *Mogibacterium*, *Blautia* and undefined genera within families BS11, Lachnospiraceae, Desulfovibrionaceae, Clostridiaceae, (Mogibacteriaceae), S24-7, Christensenellaceae, Ruminococcaceae and within the order of Clostridiales were positively correlated with ruminal proportions of isobutyric, isovaleric and total BCVFA, whereas genera *Succinivibrio*, *Prevotella*, Selenomonas and undefined genera within the families RF16 and RF32 were negatively correlated with them. Genera *Prevotella*, *Selenomonas*, *Succinivibrio*, and undefined genera within families RF16, RF32 were positively correlated, whereas genera *CF231*, (*Prevotella*), *Butyrivibrio*, *Mogibacterium*, *Blautia*, and undefined genera within families BS11, Lachnospiraceae, Desulfovibrionaceae, Clostridiaceae, (Mogibacteriaceae), S24-7, Christensenellaceae, Ruminococcaceae, and within orders of Rickettsiales and Clostridiales were negatively correlated with ruminal propionic proportions. Regarding acetate ruminal proportions, genera *Selenomonas* and undefined genera within families BS11, Ruminococcaceae, and within order Clostridiales were positively correlated, whereas genera *Succinivibrio*, *CF231*, *Pseudomonas*, and undefined genera within orders RF32 and GMD14H09 were negatively correlated.

## 4. Discussion

Setting up new circular economic approaches in livestock production by valorizing by-products of the agro-food industry as a secondary feedstuff for animal feed is crucial for the sustainability of the sector and the environment. SCG, as a by-product of the coffee industry, has been previously suggested as an alternative feed source for livestock with not very promising results [24,25,26,27]. Díaz de Otálora et al. [5] presented a new strategy of using SCG in animal nutrition. These authors formulated SCG in the concentrate at low doses, using SCG as a functional ingredient, taking advantage of some chemical constituents of SCG like coffee melanoidins and phenolic compounds which have been proposed as antimicrobials [28] and the potential beneficial effect of these active compounds on rumen microorganism and therefore ruminal fermentation. In this sense, these authors saw that SCG formulated in the concentrate at low doses improved milk yield and fat milk yield and linked this change with changes in rumen fermentation pattern. Whether the observed results were due to a change in the rumen bacterial community structure and/or changes in specific bacterial taxa and their relationship with ruminal fermentation products has been analyzed in the present study.

To our knowledge, this is the first study in analyzing the effect of increasing doses of SCG on rumen bacterial communities. We observed that the effect of SCG on microbial populations and fermentation parameters was dose-dependent for most of the measured parameters.

It could be observed that different doses of SCG linearly increased some diversity indexes (Shannon, phylogenetic diversity) probably due to an inhibiting effect of SCG on some dominant bacteria, like those of genus *Prevotella*. Moreover, some authors have proposed that SCG due to its sorption ability could stimulate the creation of microbial consortia [29] contributing to the higher bacterial diversity. These results agree with other studies on the effect of SCG on humans [30] or rat gut microbiota [31]. Pérez-Burillo et al. [30] related the increase of microbial diversity to the fiber content of SCG and especially with the mannooligosaccharides present in the SCG that could exert a prebiotic effect over some bacteria.

SCG included as a functional ingredient altered the rumen bacterial structure especially with the highest dose (100 g/kg) and also altered the relative abundance of some bacterial taxa. No previous results exist on the effects of SCG on rumen bacterial taxa, but [30] in a study performed with human feces observed enrichment in phyla Firmicutes and Bacteroidetes with SCG, and related this effect to the prebiotic effect of SCG mannooligosaccharides. These authors also observed that the increase in these two phyla was accompanied by a decrease in phylum Proteobacteria, which agrees with the results observed in the present trial. In addition, Phylum Verrucomicrobia was lower in the gut of rats fed a diet supplemented with SCG [31] in agreement with our results. At OTU level ruminal fluid of sheep in the SCG group were enriched in OTU belonging to some known fibrolytic microorganism (genera *Butyrivibrio*, *Blautia*, *Fibrobacter*, *Treponema*), which agrees with the capacity of some of these bacterial taxa to degrade lignocellulosic material present in the SCG [32] and the increased rumen acetate and butyrate proportions and the higher milk fat observed by [5]. In this sense, [30] observed that acetate and butyrate production in in vitro human gut microbial fermentation was positively correlated with the manooligosaccharides present in SCG which could have a prebiotic effect on certain bacteria. On the contrary some starch degrading bacteria genera like *Selenomonas* appeared enriched in the control group which agrees with the increased ruminal propionate proportions observed in this group by [5], as *Selenomonas ruminantium* has been commonly associated with propionic acid production in the rumen [33].

Why or how SCG induced a shift in the rumen microbiome is unknown. Spent coffee grounds contain secondary compounds, like phenolic compounds, caffeine, and tannins [28]. Also, during the roasting process of coffee, Maillard reactions occurs resulting in the formation of melanoidins, which are composed basically of sugars, amino acids, and phenolic compounds [34]. All these compounds could have had a role in the observed changes in the rumen bacterial population.

Coffee melanoidins have been proposed as antimicrobials [28], but also, it has been previously observed that melanoidins are fermented in the large intestine to produce VFA [35] acting therefore as prebiotic modulating bacterial communities. On the other hand, some authors have postulated that coffee melanoidins play a prominent role in maintaining the reducing environment in the gut [35]. This should be a matter of future assessment in the rumen, because if melanoidins have quantitative relevance in maintaining the reducing environment in the rumen, it would enhance the growth and activity of rumen fibrolytic microorganisms which agree with the results observed in the present trial.

In addition, during digestion, melanoidins and dietary fiber release some compounds [36], like phenolic acids. Some polyphenols like those present in SCG (flavonoids, chlorogenic acid, tannins, etc.) have been described as growth inhibitors for some anaerobic microorganisms [37]. Pérez-Burillo et al. [30] observed that polyphenols released by SCG hydrolysate were negatively correlated with some gut bacteria (*Blautia*, *Ruminococcus*, and *Faecalibacterium*) and that this could be related to reduced fiber degradation and VFA production. In the present work we did not observed such an effect, in fact we observed enrichment in *Blautia* when SCG were included in the concentrate, probably because in our study the content of polyphenols was more than tenfold smaller and because in the study of [30] the polyphenols were more accessible to microorganisms due to a hydrolysis process of the SCG.

Some of these phenolic compounds present in SCG, like hydroxycinnamic acids (HA) have been previously thoroughly studied. Hydroxycinnamic acids (HA), like caffeic acid, ferulic acid and chlorogenic acid between others, are phenolic compounds present in SCG [38]. Some HA like p-coumaric acid supplemented in the diet are known to limit the growth of cellulolytic bacteria [39]. In the present trial, not only did we not observe an inhibitory effect on cellulolytic bacteria, but we observed an enrichment of OTU of some cellulolytic genus like *Fibrobacter* and *Butyrivibrio* in the SCG groups. Moreover, increased acetate proportions in the rumen of SCG fed sheep was observed by [5]. In the present trial the composition of the phenolic compound present in the SCG is unknown, but the inclusion of these compounds in the total diet is notably lower than that used by [39] taking into account the SCG inclusion in the diet and the results observed in the analyses of GAE in the concentrates. On the other hand, it is known that gut microbiota and dietary polyphenols modulate each other [31]. Polyphenols are enzymatically transformed by gut microbiota, while they modify microbiota composition and functions by modulating their growth and metabolism [40]. In the rumen, microbial populations cause rapid biohydrogenation of HA followed by dehydroxylation at butyrate and more slowly at acetate [41]. At the doses present in this trial, therefore, it is possible that the ruminal microbiota and especially cellulolytic microorganisms could cope with these phenolic compounds without compromising their growth and at the same time take advantage of them. This agrees with [5] who found higher acetate and butyrate proportions in the SCG supplemented groups. Some bacterial taxa, like genera *BF331*, *Butyrivibrio*, *Blautia*, *CF231*, and undefined genera within families BS11, Ruminococcaceae, and within order Clostridiales, showed and enrichment in the SCG groups and are positively correlated with acetate and butyrate proportions in the rumen, indicating that these taxa could be implicated in this process of phenolic compounds metabolism in the rumen.

In the present study, we observed that SCG not only caused shifts in ruminal microbiota, but it can also be observed that the relationships between ruminal VFA and bacteria taxa had a little overlap between the control and the SCG groups. Moreover, in some cases certain bacterial taxa presented different correlation signs with VFA in SCG groups compared to the control group. In general correlations between VFA and bacterial taxa were sparser in the SCG groups compared to control. The positive correlation between *CF231*, *Pseudomonas*, (*Prevotella*), *Butyrivibrio*, *Mogibacterium*, *Blautia*, undefined genera within families Desulfovibrionaceae, Clostridiaceae, S24-7, Chistensenellaceae and BCVFA proportions occurred both in control and SCG groups, revealing that these bacterial taxa could have an important role in BCVFA production in the rumen. Some of these taxa were enriched in SCG groups (*Butyrivibrio*, *Blautia*, and undefined genera within family Chistensenellaceae) and could explain the increased BCVFA proportion in the SCG group found by [5]. In addition, BCVFA are essential nutrients for certain rumen microorganisms, and have been reported to enhance the growth of fiber-digesting microorganisms in the rumen [6], which agree with the results observed by [5]. Regarding acetate and propionate proportions, only genera *Selenomonas* and *Prevotella* were positively correlated with these VFA in both groups, respectively. Other microbial taxa which were not shared between groups correlated positively and negatively with these two main rumen VFA and in some cases like genus *Succinivibrio* with a different sign in the two studied groups. This may imply that acetate and propionate production is highly redundant in the ruminal microbiota, as observed previously by [42]. In fact, these two major rumen VFAs could be produced by many diverse ruminal microbes in their energy-yielding pathways [43]. The observed results, therefore, could indicate that SCG altered to some extent the correlation networks among bacterial taxa and VFA. In agreement with this, previous studies had shown that diet or dietary additives containing phenolic compounds could alter metabolic networks in the rumen [42,44,45]. However, more in-depth studies are needed to elucidate the effect of SCG on rumen microbial metabolic pathways and transcriptional patterns.

## 5. Conclusions

SCG included as a functional ingredient at low doses in the concentrate of dairy sheep increased rumen bacterial diversity and induced shifts in the relative abundance of some bacterial taxa in a dose-dependent manner. Moreover, SCG altered to some extent the correlation networks among bacterial taxa and VFA produced in the rumen, especially those related to acetic and propionic acids’ production.

## Figures and Tables

**Figure 1 microorganisms-08-01961-f001:**
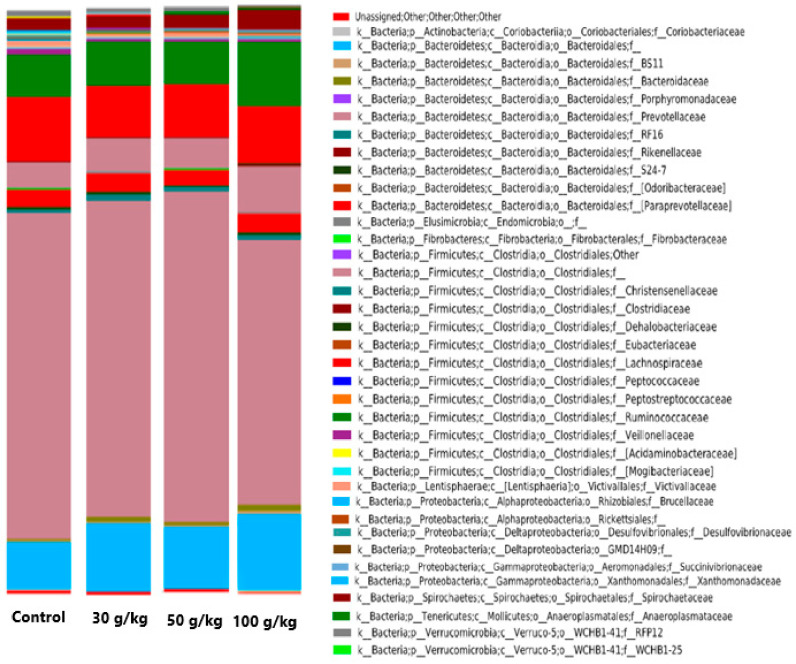
Bacterial community composition at the family level in the ruminal samples of sheep when fed a concentrate with different doses of spent coffee grounds (*n* = 9 for spent coffee grounds group and *n* = 8 for control group).

**Figure 2 microorganisms-08-01961-f002:**
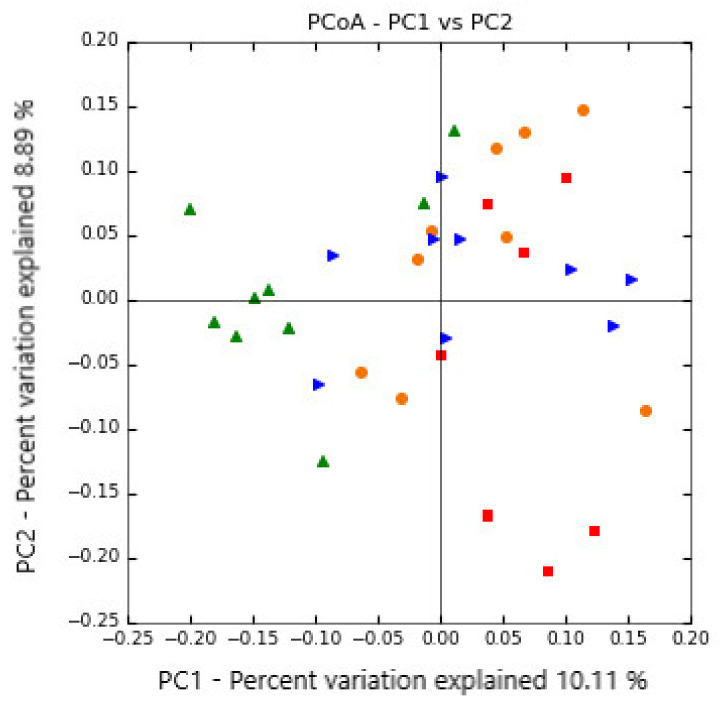
Principal coordinate analysis plot (PCoA) of the Bray-Curtis dissimilarities for the ruminal samples in the control (red square; *n* = 8), 30 g/kg spent coffee grounds (blue triangle; *n* = 9), 50 g/kg spent coffee grounds (orange circle; *n* = 9) and 100 g/kg spent coffee grounds (green triangle; *n* = 9) groups. The percentage of variation explained by each principal coordinate is indicated on the axes.

**Figure 3 microorganisms-08-01961-f003:**
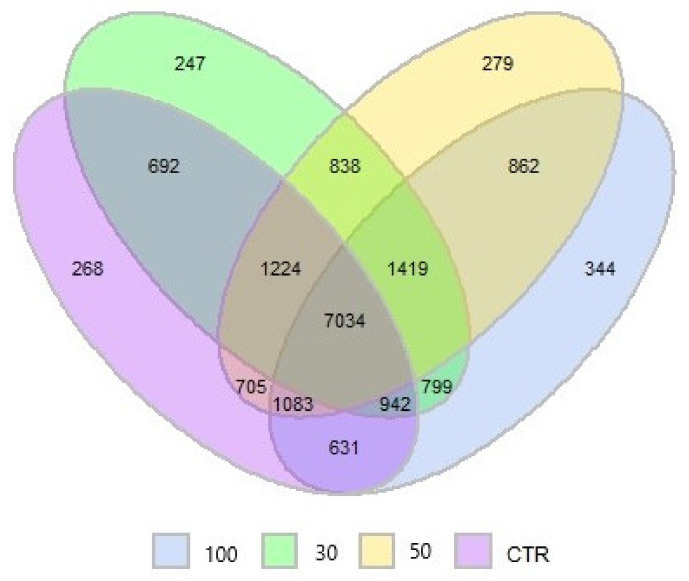
Venn diagram showing Operational taxonomic unit shared between the four treatments (CTR: control (*n* = 8); 30: 30 g/kg spent coffee grounds (*n* = 9); 50: 50 g/kg spent coffee grounds (*n* = 9); 100: 100 g/kg spent coffee grounds (*n* = 9)).

**Figure 4 microorganisms-08-01961-f004:**
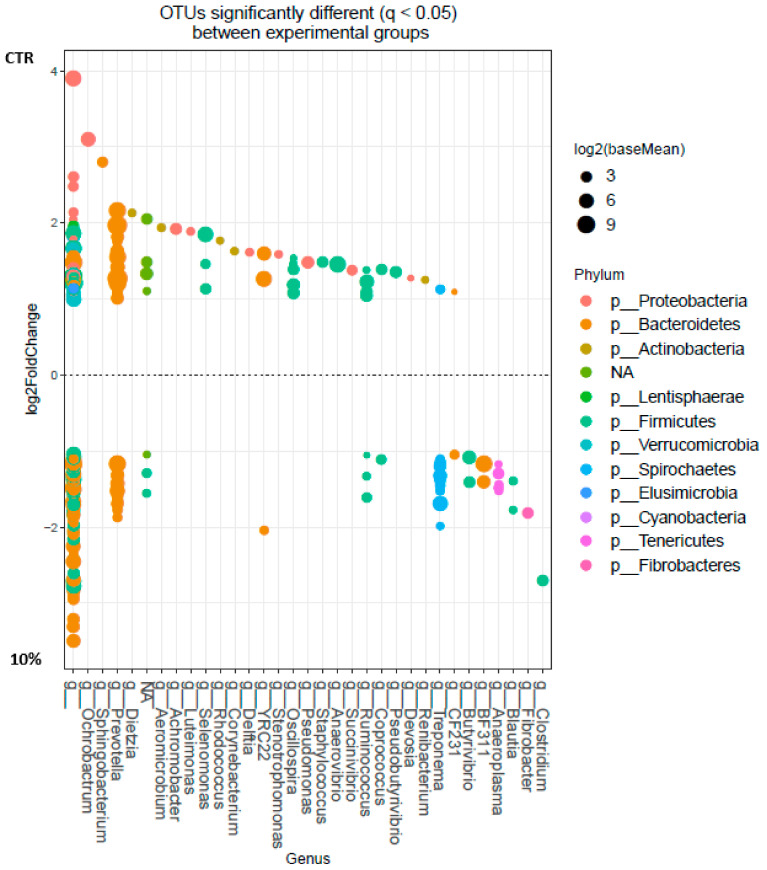
Operational Taxonomic Unit (OTU) at genus level significantly different (q < 0.05) between rumen samples of sheep fed control (above; *n* = 8) and concentrate with 100 g/kg of SCG (10%, below; *n* = 9). Each point represents a single OTU coloured by phylum and grouped on the x-axis by taxonomy, size of point reflects the log2 mean abundance of the sequence data.

**Figure 5 microorganisms-08-01961-f005:**
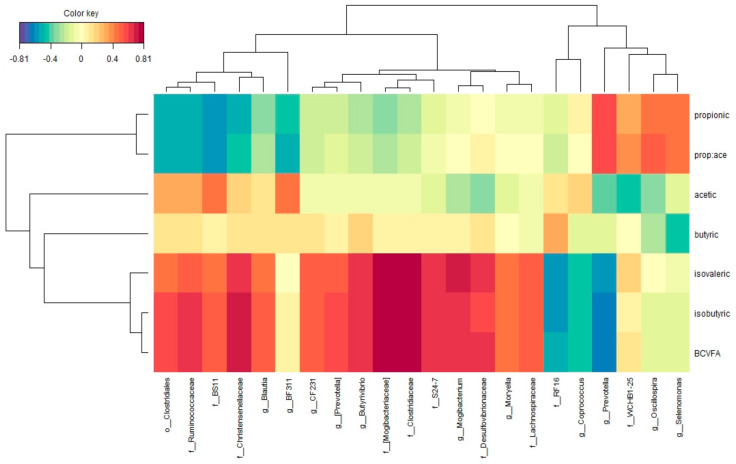
Relationships between clusters of bacterial genus and rumen VFA proportions independent of treatment (*n* = 35). This clustered image map was based on the regularized canonical correlations between relative bacterial abundances and relative concentrations of rumen volatile fatty acids. Significant correlations are coloured following the key shown.

**Figure 6 microorganisms-08-01961-f006:**
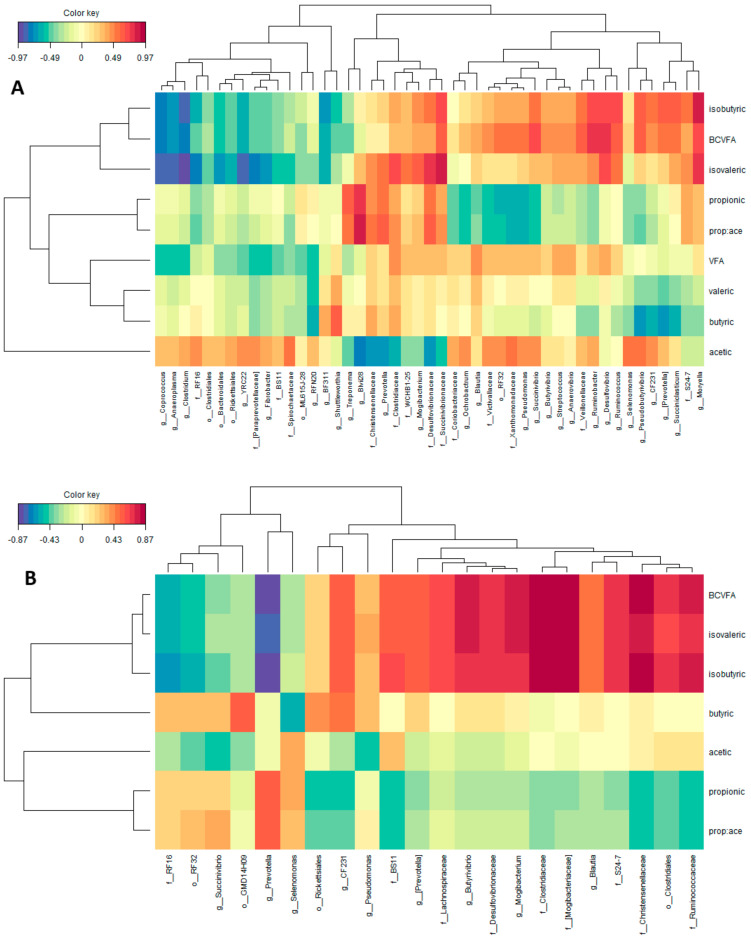
Relationships between clusters of bacterial genus and rumen VFA proportions in the Control group (**A**; *n* = 8) and in the groups fed SCG at different doses (**B**; *n* = 27). This clustered image map was based on the regularized canonical correlations between relative bacterial abundances and relative concentrations of rumen volatile fatty acids. Significant correlations are coloured following the key shown.

**Table 1 microorganisms-08-01961-t001:** Effect of different doses of spent coffee grounds on ruminal microbiota diversity measurements in lactating sheep.

Diversity Measurements	Treatment ^a^				SEM	*p*-Value	
	Control	3	5	10		Linear	Quadratic
Observed OTU	4347	4501	4659	4685	445.3	0.116	0.491
Chao1	7025	7330	7607	7650	698.8	0.069	0.393
Coverage (%)	97.8	97.5	97.4	97.4	0.35	0.034	0.278
Phylogenetic diversity	124	128	130	135	8.3	0.007	0.897
Shannon	7.82	8.33	8.33	8.72	0.4547	0.001	0.412

^a^ Control concentrate; 3, 5 and 10, treatment groups supplemented with 30 g/kg, 50 g/kg and 100 g/kg of spent coffee grounds in the concentrate, respectively; SEM: standard error of the mean.

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
