# Peer review of "Spent Coffee Grounds Alter Bacterial Communities in Latxa Dairy Ewes"

_microorganisms, 2020, doi:10.3390/microorganisms8121961_

Round 1

Reviewer 1 Report

This is a well designed paper dealing with an important subject, the structure is clear and concise and is interesting. The results section reads like a shopping list, so this should be redrafted to make it more appealing to read. Also, the resolution of the images needs to be improved. 

Line 18 DNA "was" extracted
Line 26 Not sure what the word sign means here.

The last sentence of the abstract should put the research findings in context with relation to the big picture.
Line 35 world coffee not coffee world
Line 51 "the" authors knowledge
Line 86 not clear grammar.
Line 91 use the proper celsius symbol

Section 2.5 check the paragraph formatting

Line 152 be a little more specific, statistical or technical outlier?
Line 157 importance or abundance?
Line 166, All figure legends need more detail, n numbers and statistics
Line 180 , p = or p <
The quality of the images makes it difficult to read, a higher resolution should be included
Line 159 singular (map)
Line 436 depth, not deep

Author Response

This is a well designed paper dealing with an important subject, the structure is clear and concise and is interesting. The results section reads like a shopping list, so this should be redrafted to make it more appealing to read. Also, the resolution of the images needs to be improved. 

Thank you for your valuable comments. I understand your concern about the Results section. However it is a challenge to describe these kind of results in a more appealing manner without losing information and we have not changed them. Regarding The resolution of the Images, they have been revised before the first submission and no higher resolution was achieved.

Line 18 DNA "was" extracted

Corrected

Line 26 Not sure what the word sign means here.

The sentence has been rewritten

The last sentence of the abstract should put the research findings in context with relation to the big picture.

The abstract has a limited number of words and we tried to summarize most of the important ideas on it.

Line 35 world coffee not coffee world

Corrected

Line 51 "the" authors knowledge

Corrected

Line 86 not clear grammar.

The sentence has been rewritten

Line 91 use the proper celsius symbol

Corrected

Section 2.5 check the paragraph formatting

The formatting has been checked and it is ok, the problem is that the two first sentences were very short.

Line 152 be a little more specific, statistical or technical outlier?

Corrected

Line 157 importance or abundance?

It is abundance because data are relative abundances

Line 166, All figure legends need more detail, n numbers and statistics

Figure legends have been changed and n included

Line 180 , p = or p <

It is p=

The quality of the images makes it difficult to read, a higher resolution should be included

Already answered above

Line 159 singular (map)

Corrected

Line 436 depth, not deep

Corrected

Reviewer 2 Report

The paper studied the role of spent coffee grounds (SCG) in changing the rumen bacterial community. I think it makes a solid contribution to the field. My main concern is the experiment of feeding SCG. The authors didn't mention if the SCG was sterilized before feeding. This is critical for bacterial structure analysis. The study showed that the highest difference of OTU was observed between the 100 g/kg SCG group and the control. It would be interesting to see if a higher concentration of SCG (>100g/kg)will give a higher difference of OTU and which concentration will give the highest one.

other comments:

  1. In figure 1, some colors are too similar to be distinguished.  
  2.  The authors used 'SGC' several times in the text, e.g. line 20. 'SGC increased...'.

Author Response

The paper studied the role of spent coffee grounds (SCG) in changing the rumen bacterial community. I think it makes a solid contribution to the field. My main concern is the experiment of feeding SCG. The authors didn't mention if the SCG was sterilized before feeding. This is critical for bacterial structure analysis. The study showed that the highest difference of OTU was observed between the 100 g/kg SCG group and the control. It would be interesting to see if a higher concentration of SCG (>100g/kg)will give a higher difference of OTU and which concentration will give the highest one.

Thank you for your valuable comments. The spent coffee grounds  were stabilized and dried using the flash dryer technology (RINA-JET 1008, Riera Nadeu S.A.). More in depth explanations can be found in (San Martin, D.; Orive, M.; Iñarra, B.; Garcia-Rodriguez, A.; Goiri, I.; Atxaerandio, R.; Urkiza, J.; Zufia, J. Spent coffee ground as second-generation feedstuff for dairy cattle. Biomass Conv. Bioref. 2020, https://doi.org/10.1007/s13399-020-00610-7). Briefly, this is not a sterilizing process, but they are dried in this device with an input air temperature of 330 ° C and an output air temperature of 140 ° C.

The reviewer is right and it would be interesting to analyse in the future higher doses.

other comments:

  1. In figure 1, some colors are too similar to be distinguished.  

The Reviewer is right but this is an image that the QUIIme pipeline provides. There are lots of families to show in the bar charts and sometimes colours seem very similar. Anyway they are ordered from bottom to top in the charts and can be followed with the legend.

  1.  The authors used 'SGC' several times in the text, e.g. line 20. 'SGC increased...'.

Sorry about that. SGC has been changed to SCG in line 20 and elsewhere in the manuscript